# Effective coverage of nutrition interventions across the continuum of care in Bangladesh: insights from nationwide cross-sectional household and health facility surveys

Phuong Hong Nguyen [1], Long Quỳnh Khương,[2] Priyanjana Pramanik,[1] Sk Masum Billah,[3] Purnima Menon,[1] Ellen Piwoz,[4] Hannah H Leslie [5]

For numbered affiliations see end of article.

**Correspondence to**
Dr Phuong Hong Nguyen;
P.H.Nguyen@cgiar.org

## ABSTRACT

**Introduction** Improving the impact of nutrition interventions requires adequate measurement of both reach and quality of interventions, but limited evidence exists on advancing coverage measurement. We adjusted contact-based coverage estimates, taking into consideration the inputs required to deliver quality nutrition services, to calculate input-adjusted coverage of nutrition interventions across the continuum of care from pregnancy through early childhood in Bangladesh.

**Methods** We used data from the 2014 Bangladesh Demographic and Health Surveys to assess use of maternal and child health services and the 2014 Service Provision Assessment to determine facility readiness to deliver nutrition interventions. Service readiness captured availability of nutrition-specific inputs (including human resources and training, equipment, diagnostics and medicines). Contact coverage was combined with service readiness to create a measure of input-adjusted coverage at the national and regional levels, across place of residence, and by maternal education and household socioeconomic quintiles.

**Results** Contact coverage varied from 28% for attending at least four ANC visits to 38% for institutional delivery, 35% for child growth monitoring and 81% for sick child care. Facilities demonstrated incomplete readiness for nutrition interventions, ranging from 48% to 51% across services. Nutrition input-adjusted coverage was suboptimal (18% for ANC, 23% for institutional delivery, 20% for child growth monitoring and 52% for sick child care) and varied between regions within the country. Inequalities in input-adjusted coverage were large during ANC and institutional delivery (14–17 percentage points (pp) between urban and rural areas, 15 pp between low and high education, and 28-34 pp between highest and lowest wealth quintiles) and less variable for sick child care (<2 pp).

**Conclusion** Nutrition input-adjusted coverage was suboptimal and varied subnationally and across the continuum of care in Bangladesh. Special efforts are needed to improve the reach as well as the quality of health and nutrition services to achieve the Sustainable Development Goals.

## Strengths and limitations of this study

► Our study used nationally representative population and health system data to estimate input-adjusted coverage for nutrition interventions across the continuum of maternal and early childhood care.

► Our study quantified inequities in input-adjusted coverage within the population and indicated potential priorities for improvement, including special investment in reaching underserved populations with key services, while also ensuring adequate quality of services.

► Although the Service Provision Assessment survey was quite comprehensive, it did not capture every aspect to measure facility readiness to provide nutrition interventions or to calculate each step of care cascades from need for services through to health benefits.

► The exclusion of small private clinics hospitals with less than 20 beds could lead to an overestimation of quality for private sector users, as larger facilities tend to have higher readiness than smaller ones.

## INTRODUCTION

Global evidence suggests that if high-quality health systems could effectively deliver a core subset of 19 maternal and child interventions, almost one quarter of the maternal deaths, neonatal deaths, and stillbirths would be prevented.[1] Access to 10 evidence-based nutrition interventions alone could reduce the mortality rate in children under 5 by 15%.[2] Despite this, coverage data for high-impact interventions along the continuum of care remain limited due to challenges in data collection and measurement.[3] Improving the impact of nutrition interventions requires adequate measurement of both reach and quality of interventions. Current global measurement mainly focuses on contact coverage indicators,[4] but does not adequately

capture the quality of services delivered, thus providing only weak links with actual health benefits received by the population in need.[5]

Effective coverage has been variously defined as 'the fraction of maximum possible health gain an individual with a healthcare need can expect to receive from the health system'[6] and as a situation where 'people who need health services obtain them in a timely manner and at a level of quality necessary to obtain the desired effect and potential health gains.'[7] Effective coverage goes beyond contact coverage measurement by using not just the receipt of services by clients but also the quality of care received. Multiple quality domains can be considered, including inputs (service and equipment availability, training, etc), service delivery processes (eg, compliance with protocols and standards of care) and outcomes (health benefits, client satisfaction etc).[8] However, different studies have used various definitions and measurements for effective coverage, thus limiting the comparability among studies. To address this challenge, the recently proposed seven-step coverage framework begins with the target client population, and following them through a hypothetical cascade of the losses of health benefits at different stages, including service contact (an individual in need contacts health service), input-adjusted coverage (contacts health service that is ready), intervention coverage (receives health services), quality-adjusted coverage (receives health service according to standard), user adherence-adjusted coverage (user adherence) and outcome-adjusted coverage (health gain achieved).[5 9]

Effective coverage has been found to be considerably lower than contact coverage for antenatal care (ANC),[10 11] family planning and sick child care in several low-income and middle-income countries (LMICs).[11] Findings from LMICs also suggested low levels of availability, readiness and coverage of obstetric services[12] and showed substantial reductions (20–39 percentage points (pp)) from contact coverage to effective coverage in facility birth.[13] Overall, a review of 36 studies showed large gaps between contact coverage and quality-adjusted coverage levels (10–38 pp) across the continuum of care for reproductive, maternal, newborn and child health.[5]

Although recent studies have paid more attention to capturing the quality of care and quantifying the gaps between contact coverage and effective coverage across the continuum of care, studies on nutrition-related interventions are scarce. Previous studies mainly came from small-scale studies that focused on fortified foods where contact coverage was measured through exposure to a particular fortified food and quality-adjusted measures were based on regular consumption of the fortified food.[5] Recently, one study in Malawi showed that even though the utilisation of ANC and facility delivery was high, women often did not receive nutrition-related interventions (such as iron interventions, nutrition counselling and breastfeeding counselling).[14]

Bangladesh has observed significant improvement in contact coverage in maternal and child health in the last decade,[15] yet substantial gaps remain in coverage of nutrition services, with less than 30% of women receiving all four key ANC nutrition interventions—iron folic acid (IFA) supplement, calcium supplements, dietary advice and weight monitoring, and only 25% of sick children having their weight checked against a growth chart.[16] Large gaps in facility readiness to provide nutrition interventions persist. Specifically, only around half of providers reported having received any training on nutrition, and only 13% of facilities possessed more than 5 of the 13 items deemed essential for service delivery for children.[16] Previous studies in Bangladesh have reported separately on contact coverage[17–19] and quality of services provided.[16 20] This paper aimed to adjust contact-based health coverage estimates in Bangladesh, taking into consideration the inputs required to deliver quality nutrition interventions across the continuum of care, specifically ANC and delivery for women and growth monitoring and curative care for young children.

## DATA AND METHODS
### Data sources
We used two sets of publicly available nationally representative data: the 2014 Bangladesh Demographic and Health Survey (DHS)[21] and the 2014 Service Provision Assessment (SPA).[22] The DHS survey was a population-based household surveys that provided representative data on health and nutrition indicators at national and regional levels.[21] The survey was based on a two-stage stratified sample of households. In the first stage, 600 enumeration areas were selected with probability proportional to the enumeration area size. In the second stage of sampling, a systematic sample of 30 households on average was selected from a household list in each selected cluster by equal probability sampling. All women in selected households with a birth in the 3 years before the survey were interviewed about antenatal care and birth care for the most recent live birth. Child curative care questions were asked for all living child under 5 years in the household.

The SPA is a formal health facility-based survey designed to provide information on health service availability and readiness, with an emphasis on the delivery of reproductive, maternal, newborn and child health services.[22] The Bangladesh SPA included a standard set of survey instruments for a facility inventory and a healthcare provider interview. Facilities were assessed for the availability of health services, basic amenities, equipment, laboratory services, essential medicines and human resources. The SPA survey captures tracer indicators to monitor health system strengthening.[23] Health facilities were sampled from a complete listing of 19 184 registered health facilities in the country, with the aim to provide the survey results separately for the seven administrative divisions and different types of public facilities, non-governmental organisation (NGO) clinics/hospitals and private hospitals. Small private clinics with less than 20 beds were excluded. Further details on the survey's sampling strategy

were presented in the survey report.[22] The Bangladesh SPA did not include observation of selected client visits or exit interviews.

## Statistical analysis

Input-adjusted coverage for nutrition intervention was calculated among individuals in need of care as the product of the coverage of services and the readiness of health facilities to provide quality care.

### Coverage of services

We estimated coverage of services based on DHS data, accounting for the DHS sampling weights. Coverage of ANC care was defined as the percentage of women 15–49 years old with at least one live birth in the 3 years preceding the survey who reported at least 4 ANC visits (one of those visits from a trained provider) for their most recent birth. Coverage of institutional delivery was defined as the percentage of women with a live birth in the 3 years preceding the survey who gave birth in a health facility. There was no consensus on coverage measurement for child growth monitoring. However, growth monitoring activities, such as height, weight and mid-upper arm circumference measurement, are intended to be a core part of the Integrated Management of Childhood Illness (IMCI) protocol that is followed during sick child contacts,[16] which has been shown to have an impact on childhood stunting.[24] Based on this, and given the lack of survey questions that explicitly ask respondents about growth monitoring services, coverage of growth monitoring was defined as the percentage of all children under 5 years of age who sought care for illness at a formal health facility. Care-seeking for well child visits is not captured in the DHS; moreover, delivery of growth monitoring services in Bangladesh is delivered via IMCI, so while this indicator provides an incomplete picture of growth monitoring, it does correspond to the current service delivery model. Coverage of sick child care was defined as the percentage of children with any illness who sought care at a formal health facility.

### Facility readiness to provide nutrition interventions

We analysed facility readiness to deliver nutrition interventions based on five attributes as guided by WHO's Service Availability and Readiness Assessment Manual[25] including: (1) trained personnel, (2) guidelines, (3) equipment, (4) diagnostic capacity and (5) medicines. We used the tracer indicators that are most relevant to capture selected nutrition interventions. The selected indicators for estimating input-adjusted coverage for nutrition interventions across continuum of care are presented in table 1. During ANC, multiple nutrition interventions should occur; these include delivering IFA supplementation, weight monitoring for pregnant women, anaemia management, blood pressure management, fetal growth monitoring and health and nutrition education. During birth, the main nutrition intervention is support for early breastfeeding practices. During

early childhood, national protocols are for children to be weighed and to receive nutrition education, vitamin A supplementation, deworming, anaemia management and oral rehydration salts and zinc for diarrhoea.

We calculated the composite readiness score using a weighted additive approach, which weights the five domains equally within each measure. Previous studies have found that three methods (simple additive, weighted additive or principal component analysis) for calculating quality indices yield generally comparable results.[26] The weighted additive method has been recommended[27] as easy to calculate and interpret. The domain scores were calculated by averaging the indicators within each domain; domain scores were then summed and multiplied by 100 to create the total weighted additive scores which expressed as percent of total readiness. The calculated readiness score for a specific facility category was an average score of all facilities in the same category. We accounted for the SPA sampling weight when summarising readiness scores.

### Estimating input-adjusted coverage

We linked health facility and household survey data at the aggregate level, using an ecological linking method which has been shown a feasible and valid approach for estimating quality-adjusted effective coverage.[28] We disaggregated the coverage by type of facility where services were sought. In case women went to more than one type of facility for ANC, we used the highest level of facility for analyses. Facility types were harmonised between the DHS and SPA at five levels: district and upazila public facilities, union-level public facilities, public community clinic, NGO clinic/hospital and private clinic/hospital (online supplemental table S1).

Input-adjusted coverage was estimated at both the national and regional level (seven divisions), accounting for four types of facilities where the care was sought. At the regional level, the input-adjusted coverage was the summation of input-adjusted coverage of each type of facility that was constructed as the product of the coverage and readiness estimates:

$$IACr = \sum_j C_{rj} * Q_{rj}$$

where $IAC_r$ represented input-adjusted coverage in region r,

$C_{rj}$ was the proportion of women/children who sought care in facility type j in region r,

And $Q_{rj}$ was the average readiness score of facility type j in region r.

The national input-adjusted coverage was the summation of regional input-adjusted coverage weighted by the proportion of users in each region:

$$IAC_T = \sum_r IAC_r * w_r$$

where $w_r$ was the proportion of users in region r.

The variance of input-adjusted coverage estimates was calculated with an approximation procedure referred to

**Table 1** Definition of indicators for estimating input-adjusted coverage for nutrition interventions across continuum of care

| Services | Associated nutrition interventions | Measure of need (denominator) | Measure of use (numerator) | Quality estimator |
|---|---|---|---|---|
| Antenatal care | ▶ IFA supplementation<br>▶ Weight monitoring for pregnant women<br>▶ Anaemia management<br>▶ Blood pressure management<br>▶ Foetal growth monitoring<br>▶ Health and nutrition education | Women 15–49 years old with at least one child under 3 years. | Women 15–49 years old with at least one child under 3 years, whom for their most recent birth, reported at least 4 ANC visits (ANC4+). | Human resources<br>▶ Staff with any training on ANC<br>Guidelines.<br>▶ National guidelines for ANC, visual aids for client education.<br>Basic equipment<br>▶ Adult weighing scale.<br>▶ Tape measure for fundal height.<br>▶ Blood pressure apparatus.<br>▶ Stethoscope.<br>▶ Foetal stethoscope.<br>Diagnostic capacity.<br>▶ Haemoglobin.<br>▶ Urine protein.<br>Essential medicines.<br>▶ Iron tablets.<br>▶ IFA tablets. |
| Birth care | ▶ Child birth weight measures<br>▶ Early breastfeeding practices | Women 15–49 years old with at least one child under 5 years | Women 15–49 years old with at least one child under 5 years, whom for their most recent birth, reported delivery in a health facility | Human resources<br>▶ Staff with any training on IMPACT<br>Guidelines<br>▶ Guidelines on basic birth care BEmONC<br>▶ Guidelines on comprehensive birth care: CEmONC<br>Basic equipment<br>▶ Infant scale<br>▶ Manual or digital BP apparatus |
| Child growth monitoring | Child growth monitoring | All children alive between 0 and 59 months | All children who had diarrhoea or ARI symptoms for whom care was sought from a medical provider | Human resources<br>▶ Staff with any training on growth monitoring<br>Guidelines<br>▶ Guidelines for growth monitoring<br>Basic equipment:<br>▶ Scale<br>▶ Length or height board<br>▶ Tape for measuring head<br>▶ Growth chart |
| Sick child care | ▶ Nutrition education<br>▶ Vitamin A supplementation<br>▶ Deworming<br>▶ Anaemia management<br>ORS and zinc for diarrhoea | All children alive between 0 and 59 months who had diarrhoea or ARI in the last 2 weeks | All children who had diarrhoea or ARI symptoms for whom care was sought from a medical provider | Human resources<br>▶ Staff with any training on IMCI<br>Guidelines<br>▶ IMCI guideline: national guidelines for IMCI, IMCI chart booklet, IMCI card, other visual aids<br>Basic equipment<br>▶ Scale<br>Diagnostic capacity<br>▶ Haemoglobin<br>Essential medicines:<br>▶ ORS<br>▶ Albendazole/ mebendazole<br>▶ Iron tablet<br>▶ Vitamin A<br>▶ Zinc tablet/zinc sulphate syrup. |

ANC, antenatal care; ARI, acute respiratory infection; BEmONC, Basic Emergency Obstetric and Neonatal Care; BP, blood pressure; CEmONC, Comprehensive Emergency Obstetric and Neonatal Care; IFA, iron folic acid; IMCI, Integrated Management of Childhood Illness; IMPACT, Integrated management of pregnancy and childbirth; ORS, oral rehydration salts.

as the 'delta' method which has been used and described in detail in previous research.[29] Finally, we calculated the input-adjusted coverage by strata of place of residence, educational level and socioeconomic status (SES) quintiles. This analysis assumed that there was no systematic difference in facility readiness within region by the

education or socioeconomic stratifiers (eg, that within a given subnational region, the average readiness of public community clinics used by highly educated women was the same as average readiness of public community clinics used by less educated women). Data analysis was performed using Stata V.16.0.

## Patient and public involvement

Patients and/or the public were not involved in the design, or conduct, or reporting, or dissemination plans of this research.

## RESULTS

The household survey included 17 863 women (out of 18 245 eligible women aged 15–49 years, a response rate of 98%), among them 4488 women reporting a birth in the 3 years preceding the survey and 7886 women with living children under 5 years. Health service coverage varied considerably between services, with low coverage for ANC4+ (28%), institutional delivery (38%), child growth monitoring (35%), but substantially higher for sick child care (81%). The place of ANC and childcare by facility type was shown in online supplemental figure S1.

The health facility survey included 1548 facilities of 1596 sampled (97% response rate); among them 1493 facilities (96%) offering services on antenatal care, 586 (38%) offering services on delivery care, and 1463 (92%) offering curative care for sick children. Government health facilities are the major providers of care (online supplemental figure S2).

The availability of tracer indicators that would be required to deliver interventions across the continuum of care is presented in table 2 for the national level. Overall, facilities lacked trained personnel, guidelines, equipment, medicines or supplies needed to provide high quality of care (table 2). Of facilities offering each service, approximately half of facilities had staff with any training on ANC, child growth monitoring, and IMCI, but a lower proportion of facilities (39%) had staff with any training on integrated management of pregnancy and childbirth (IMPAC). Similarly, around half of facilities had national or other guidelines for ANC, growth monitoring or IMCI, but only 27% had guidelines for Basic Emergency Obstetric and Neonatal Care or Comprehensive Emergency Obstetric and Neonatal Care. While a majority of facilities (>85%) had adult weighing scales, blood pressure monitors and stethoscopes, only 36% had tape measures for fundal height and 15% had fetal stethoscopes. Child scales, length or height board and growth chart were only available in 60%–70% of facilities that offered sick-child care. Regarding diagnostic capacity, 11% of facilities were able to test for haemoglobin and 18% for urine protein. IFA supplements were available in 91% of facilities. Zinc or vitamin A supplementation was reported in three quarters of facilities and deworming for children was reported in almost all facilities. The availability of these items varied

by division (online supplemental table S2) and by types of facilities (online supplemental table S3).

Readiness scores were calculated at the national (table 2) and regional levels and by facility type (figure 1). Overall, facilities demonstrated 51% of the expected resources to provide nutrition interventions during ANC, 48% for delivery care, 51% for growth monitoring, and approximately half the required resources for child health services. Readiness scores varied by regions and depending on the service. Rangpur performed the best overall, and Sylhet showed high readiness for ANC but lowest on delivery care and average on others (figure 1). District and upazila public facilities had the highest readiness scores across the continuum of care, whereas lower-level facilities (union level and public community) were much less prepared. For example, district public hospitals had the highest readiness score, with 76% of the expected capacity to provide nutrition interventions during ANC care services, but public community clinics had a readiness score of only 49%.

Figure 2 plots health service coverage against facilities' readiness score in each region of the country. For ANC4+, the regional averages of coverage and readiness fell mostly in quadrant III for all regions, indicating both low coverage (ranged between 18% and 36%) and moderate readiness (between 48% and 57%) (table 3), thus, the priority is improving coverage as well as quality of nutrition interventions for ANC care. Similar priority was observed for delivery care: most regions had low access (30%–40% except Khulna) and low readiness (less than 50% except Rangpur). For child growth monitoring, all regions showed low coverage (30%–39%); Dhaka had the lowest readiness (45%) while other regions had a higher but wide range of readiness (50%–60%). For sick child care, regions had similar levels of high coverage (>75%) but showed lower and highly variable readiness.

Taking into account the readiness of facilities to provide nutrition interventions and the use of services, we estimated the input-adjusted coverage at the national and regional level (table 3 and figure 3). At the national level, input-adjusted coverage was 18% for ANC4+, 23% for delivery care, 20% for child growth monitoring and 52% for sick child care. The gaps between contact and input-adjusted coverage ranged between 10 and 30 pp.

When examining the inequality gaps, we found that input-adjusted coverage varied by as much as 6–32 pp between regions within the country (figure 3). Inequality in input-adjusted coverage was large during pregnancy and delivery (14–17 pp between urban and rural areas, 15 pp between low and high education, and 28–34 pp between the rich and the poor), but narrower for child growth monitoring and sick child care (<2 pp) (figure 4).

## DISCUSSION

Improving both coverage and quality of healthcare services is essential to achieving universal health coverage and achieving the third Sustainable Development Goal

**Table 2** Percentage of health facilities with structural input items and their readiness score

| | Input items | % facilities with item available | 95% CI |
|---|---|---|---|
| **Antenatal care (n=1493 facilities)** | | | |
| Staff | Staff with any training on ANC | 48.75 | 43.24 to 54.26 |
| Guideline | National or other guidelines for ANC | 49.64 | 41.88 to 57.40 |
| | Visual aids for client education | 66.93 | 60.71 to 73.15 |
| Functional equipment | Adult weighting scale | 85.44 | 81.83 to 89.06 |
| | Tape measure for fundal height | 36.17 | 29.37 to 42.97 |
| | Blood pressure apparatus | 91.22 | 88.21 to 94.24 |
| | Stethoscope | 91.93 | 89.06 to 94.79 |
| | Foetal stethoscope | 15.25 | 10.77 to 19.74 |
| Diagnostics capacity | Haemoglobin | 10.57 | 7.15 to 13.99 |
| | Urine protein | 17.87 | 13.62 to 22.12 |
| Medicine | Iron tablets | 60.06 | 51.75 to 68.37 |
| | IFA tablets | 83.88 | 79.02 to 88.73 |
| **Readiness score for ANC** | | **51.44** | **48.56 to 54.32** |
| **Birth care (n=586 facilities)** | | | |
| Staff | Staff with any training on IMPAC | 38.74 | 30.01 to 47.47 |
| Guideline | Guidelines for BEmONC or CEmONC | 27.18 | 17.28 to 37.09 |
| Functional equipment | Infant scale | 60.64 | 52.62 to 68.65 |
| | Manual or digital BP apparatus | 93.35 | 89.08 to 97.62 |
| **Readiness score for birth care** | | **47.64** | **41.81 to 53.46** |
| **Child growth monitoring (n=1463 facilities)** | | | |
| Staff | Staff with any training on growth monitoring | 48.76 | 42.46 to 55.07 |
| Guideline | Guidelines for growth monitoring | 46.65 | 39.28 to 54.01 |
| Functional equipment | Child scale (child or infant) | 69.00 | 63.08 to 74.92 |
| | Length or height board | 59.16 | 51.64 to 66.67 |
| | Tape for measuring head | 30.67 | 24.48 to 36.86 |
| | Growth chart | 66.70 | 60.66 to 72.73 |
| **Readiness score for child growth monitoring** | | **50.60** | **46.86 to 54.34** |
| **Sick child care (n=1463 facilities)** | | | |
| Staff | Staff with any training on IMCI | 53.85 | 47.36 to 60.33 |
| Guideline | Guidelines for IMCI | 50.84 | 44.29 to 57.39 |
| | IMCI chart booklet | 53.35 | 47.13 to 59.56 |
| | IMCI mother's cards (IMCI card) | 39.48 | 33.50 to 45.45 |
| | Other visual aids for teaching caretakers | 56.23 | 50.36 to 62.09 |
| Functional equipment | Child scale (child or infant) | 60.80 | 54.16 to 67.44 |
| Diagnostics capacity | Haemoglobin | 10.34 | 6.85 to 13.84 |
| Medicine | ORS | 63.59 | 58.14 to 69.04 |
| | Albendazole/mebendazole | 87.59 | 83.68 to 91.50 |
| | Iron tablet | 54.35 | 46.15 to 62.56 |
| | Vitamin A | 66.52 | 60.60 to 72.44 |
| | Zinc tablet/zinc sufate syrup | 65.75 | 59.92 to 71.57 |
| **Readiness score for sick child care** | | **48.30** | **44.90 to 51.69** |

ANC, antenatal care; BEmONC, Basic Emergency Obstetric and Neonatal Care; BP, blood pressure; CEmONC, Comprehensive Emergency Obstetric and Neonatal Care; IMCI, integrated management of childhood illness; IMPAC, integrated management of pregnancy and childbirth; ORS, oral rehydration salts.

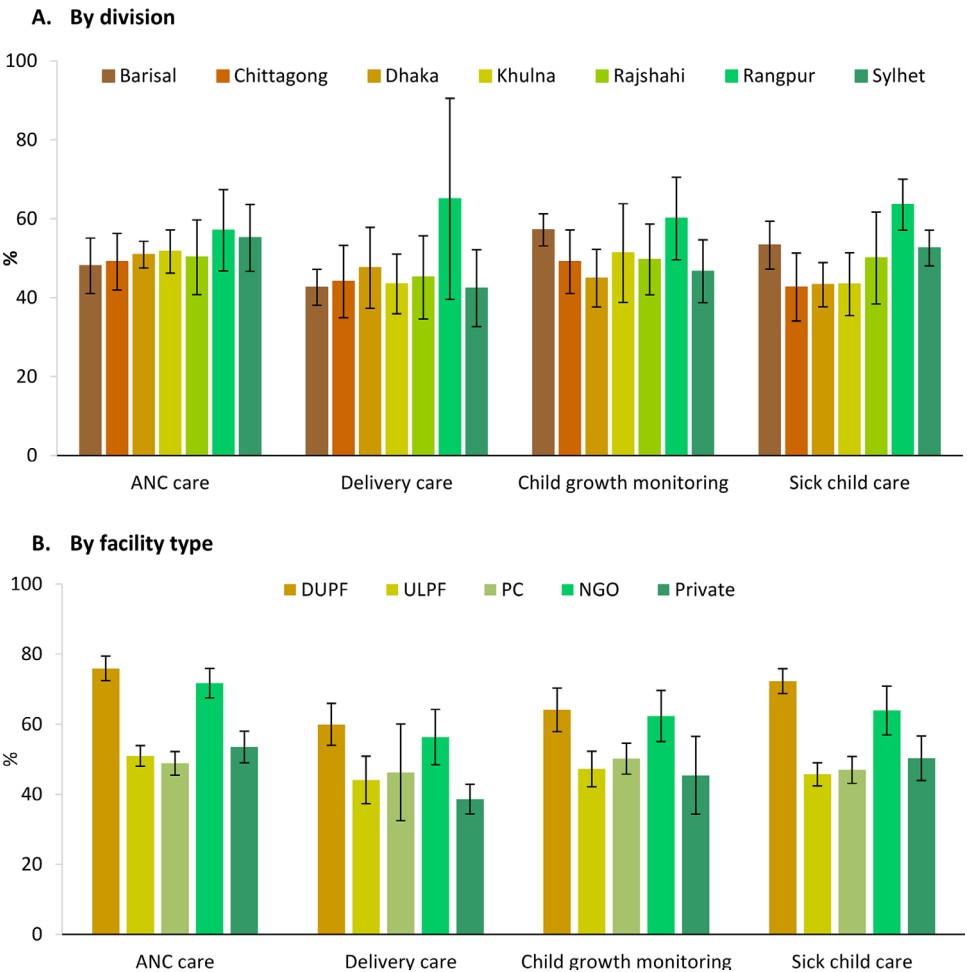

**Figure 1** Readiness score of facilities to provide nutrition interventions, by division (A) and types of facility (B), Bangladesh 2014.[1] (A) By division (B) by facility type.[1] ANC, antenatal care; DUPF, district and upazila public facilities; NGO, non-governmental organisation; PC, public community clinic; ULPF, union-level public facilities.

(SDG) of ensuring healthy lives and promoting well-being for all at all ages.[30] Our study is one of very few to develop and apply nutrition-related metrics of effective coverage across the continuum of care by incorporating service readiness of nutrition interventions into coverage estimates of ANC and postdelivery care. Our findings showed low levels of coverage across the continuum of care (28%–38%), except for sick child care (81%); we also found moderate levels of facility readiness to provide nutrition interventions (42-65). After adjustment for readiness, the input-adjusted coverage was found to be 10–30 pp lower than the contact coverage, and there was high level of inequality between regions within the country, residential areas, education and SES for ANC and birth care, but less so for sick child care. Because our estimates focused on inputs to care and did not capture how often these inputs were provided in adherence with evidence-based protocols, they should be considered an upper limit of the quality of nutrition care actually received.

The large gap between contact and adjusted coverage in our study was aligned with a review of 36 studies finding a range of 10–38 pp reductions between crude coverage and quality-adjusted coverage across the continuum of

care.[5] These findings indicated that women who sought care in a health facility did not necessarily receive the quality of care needed for them and their children. Our findings provided insight into health system challenges and indicate potential priorities for improvement.

For ANC, the priority is ensuring that all four contacts are provided by a trained provider, while improving coverage and quality simultaneously to reach high levels of effective coverage for nutrition interventions. We found a low contact coverage of 4+ ANC (28%), with evidence of inequities by wealth, education and place of residence. Evidence from a systematic review showed that both demand-side (such as maternal education, spousal education, marital status, women's employment, autonomy, cultural beliefs, media exposure and household income) and supply-side factors (such as service availability and cost) influence ANC uptake.[31 32] Financial and transport constraints may be a greater barrier for women in remote and poorer areas.[33 34] Interventions such as mobile health technologies have been found to improve ANC attendance,[35] but improving coverage requires targeted investments to reach underserved communities. We also found low readiness of the health system to provide nutritional

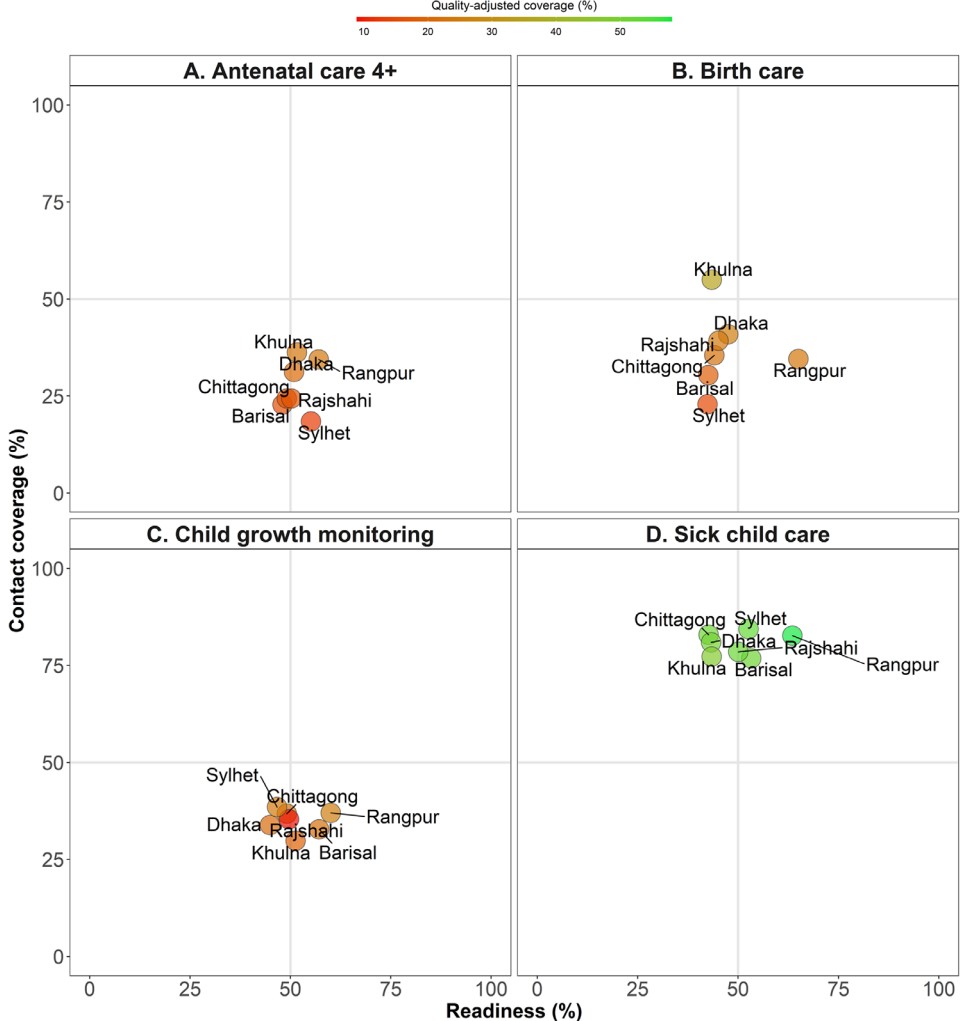

**Figure 2** Regional readiness score of facilities to provide nutrition interventions versus coverage, Bangladesh 2014.

interventions during ANC (51%) which is consistent with a previous study in Bangladesh.[16] Our findings suggested that facility readiness can be improved by ensuring the availability of trained providers and necessary equipment, as well as strengthening diagnostic capacity.

For birth care, the priority clearly lies in improving coverage of facility-based deliveries while also improving readiness and provider compliance with labour management protocols to increase support for early initiation of breast feeding. Although the trend of institutional delivery has increased over time from 29% in 2011[36 37] to 38% at the time of the survey in 2014, nearly two-thirds of all births still take place at home. Women's educational interventions and community-level mobilisation may increase the coverage of institutional delivery.[38 39] In terms of quality, facilities have low readiness (48%) to provide nutrition interventions, particularly activities related to early initiation of breast feeding, thus additional efforts are needed to improve the overall quality of service for facility-based delivery, including training staff, providing guidelines, emergency transport and necessary equipment.[40]

For care for children under 5, the priority is improving both coverage and quality of nutrition interventions for child growth monitoring. While community-based management of acute malnutrition activities such as screening and growth monitoring has been prioritised under the Bangladesh National Nutrition Services,[41] there is little data on the coverage and uptake of these interventions. Growth monitoring activities are part of the IMCI protocol that is to be followed during sick-child contacts, but implementation is poor[16] and their reach across the population is limited by the reach of public health facilities and by inclusion in sick child care rather than in well-child care.[42] This suggests the need for widely available services to deliver preventive nutrition services, not only to identify malnourished children, but to detect growth faltering early through regular growth monitoring and counselling for all children. Coverage can be increased by generating community awareness about the need for these services and generating demand. To ensure quality, trained staff should be available and have the necessary equipment to take anthropometric measurements, correctly use growth charts and counsel caregivers.

**Table 3** Estimated regional and national input-adjusted coverage of nutrition interventions across continuum of care, Bangladesh 2014

| | Contact coverage | Readiness score | Input-adjusted coverage | |
| --- | --- | --- | --- | --- |
| | | | Estimate | 95% CI |
| **Antenatal care 4+** | | | | |
| Barisal | 22.79 | 48.04 | 14.72 | 11.02 to 19.38 |
| Chittagong | 24.35 | 49.09 | 15.80 | 12.85 to 19.28 |
| Dhaka | 31.19 | 50.90 | 19.96 | 16.59 to 23.82 |
| Khulna | 36.21 | 51.65 | 23.20 | 19.01 to 27.99 |
| Rajshahi | 24.36 | 50.24 | 16.06 | 12.49 to 20.42 |
| Rangpur | 34.44 | 57.05 | 23.35 | 18.33 to 29.24 |
| Sylhet | 18.44 | 55.13 | 12.25 | 8.74 to 16.92 |
| Overall | **28.06** | **51.44** | **18.23** | **16.59 to 19.99** |
| **Birth care** | | | | |
| Barisal | 30.38 | 42.60 | 17.70 | 13.07 to 23.53 |
| Chittagong | 35.49 | 44.08 | 21.87 | 17.83 to 26.53 |
| Dhaka | 40.91 | 47.56 | 24.82 | 20.46 to 29.76 |
| Khulna | 54.96 | 43.47 | 34.96 | 26.76 to 44.17 |
| Rajshahi | 39.23 | 45.13 | 23.57 | 19.38 to 28.34 |
| Rangpur | 34.52 | 65.02 | 22.89 | 17.70 to 29.07 |
| Sylhet | 22.87 | 42.39 | 14.12 | 10.79 to 18.25 |
| Overall | **37.73** | **47.64** | **23.24** | **21.11 to 25.51** |
| **Child growth monitoring** | | | | |
| Barisal | 32.77 | 57.17 | 21.79 | 17.33 to 27.01 |
| Chittagong | 36.78 | 49.10 | 22.57 | 19.45 to 26.03 |
| Dhaka | 33.87 | 44.92 | 20.47 | 17.84 to 23.37 |
| Khulna | 29.79 | 51.30 | 18.27 | 14.97 to 22.12 |
| Rajshahi | 35.34 | 49.68 | 8.94 | 7.66 to 10.41 |
| Rangpur | 36.99 | 60.06 | 24.93 | 17.99 to 33.45 |
| Sylhet | 38.51 | 46.66 | 23.64 | 16.39 to 32.84 |
| Overall | **35.05** | **50.60** | **20.41** | **18.76 to 22.16** |
| **Sick child care** | | | | |
| Barisal | 76.83 | 53.28 | 50.95 | 41.64 to 60.19 |
| Chittagong | 82.92 | 42.65 | 51.31 | 41.50 to 61.02 |
| Dhaka | 80.95 | 43.29 | 50.50 | 42.42 to 58.56 |
| Khulna | 77.28 | 43.43 | 47.66 | 38.13 to 57.36 |
| Rajshahi | 78.53 | 50.04 | 51.92 | 42.02 to 61.68 |
| Rangpur | 82.69 | 63.54 | 58.02 | 48.95 to 66.59 |
| Sylhet | 84.38 | 52.60 | 52.69 | 40.21 to 64.84 |
| Overall | **81.20** | **48.30** | **51.67** | **47.56 to 55.76** |

Measurement of care seeking among well children would help in monitoring progress in this area. Other elements of overall sick-child care quality should be improved by ensuring that staff are trained on IMCI, and that guidelines, equipment and medicine are available.

The inequality in adjusted coverage highlights targets for intervention according to administrative division and populations served. There is a clear variability in the adjusted coverage of indicators across administrative divisions, with some divisions (Khulna and Rangpur) performing better overall and others (Sylhet) performing poorly overall. For Sylhet specifically, poor performance may be because of its terrain, which makes many areas hard to reach. However, lessons should be shared from the successes of high-performing divisions such as Khulna and Rangpur, and focused strategic planning and action

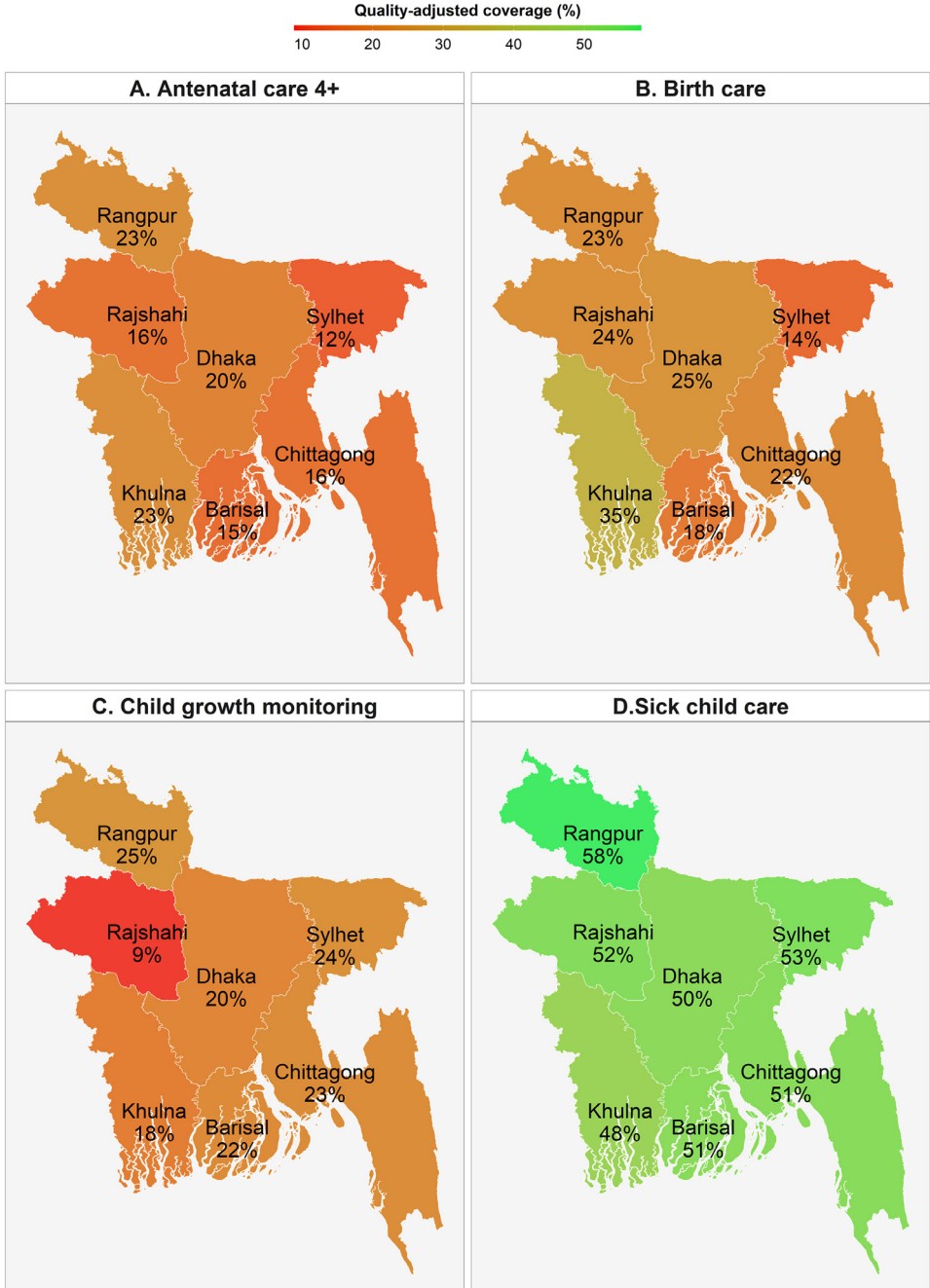

**Figure 3** Input-adjusted coverage of nutrition interventions by region, Bangladesh 2014.

could help reduce regional disparities. Input-adjusted coverage also showed inequities by place of residence, education and SES. However, the size of these inequalities varied where the gaps were larger in the case of ANC and delivery, but smaller for child growth monitoring and sick-child care. Our findings of inequalities in coverage during pregnancy and delivery align with previous research finding that urban residence, higher education attainment (for the respondent and spouse), and higher wealth status significantly predict utilisation of institutional delivery services[37] and 4+ ANC exposure.[17] Thus, there is a special need to specifically target underserved populations.

Investing in strengthening effective coverage of nutrition interventions is particularly important for a resource-poor country like Bangladesh which still carries a large burden of poor maternal and child health and nutrition. Despite consistent improvements in coverage of key nutrition service delivery platforms/contacts that is, ANC and child health services, the health sector can do much more to strengthen nutrition service delivery in Bangladesh. The rapid growing private sector in the context of an ineffective regulatory environment has contributed to the lack of standardised practice in this setting.[43] Investing in the quality of public services, therefore, could bring clients back to use public services and ensure better coverage

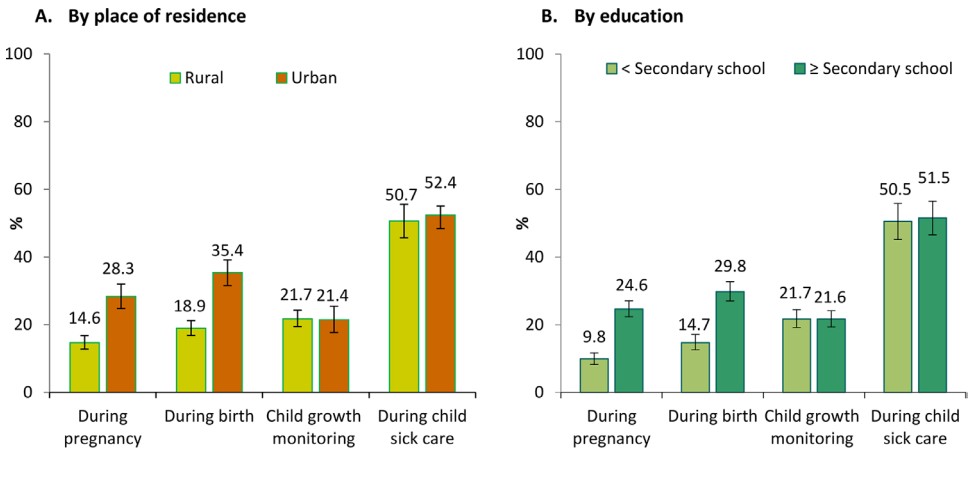

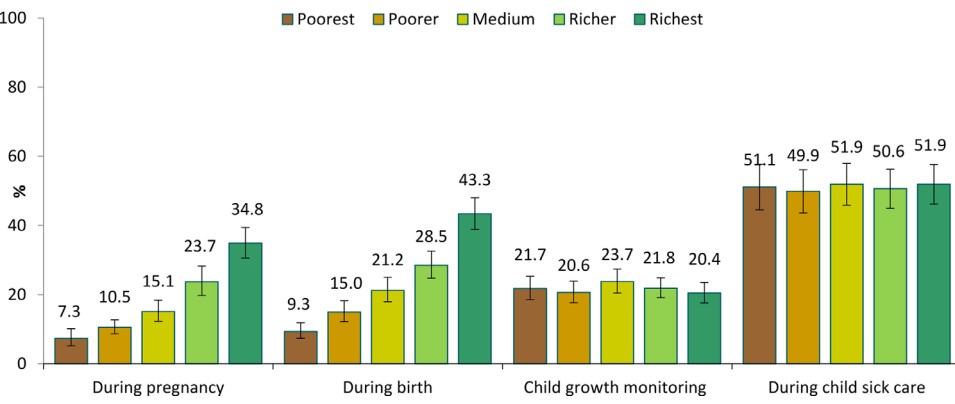

**Figure 4** Inequality in input-adjusted coverage of nutrition interventions, Bangladesh 2014.

of higher-quality care. The challenge for governments scaling proven interventions at desired coverage and quality in such settings, however, is multidimensional. Depending on the type of contact and the intervention, it is likely that investments in strengthening the public sector platforms require both structural and process fixes. In concert with this, the government's stewardship in establishing functioning regulatory and accountability mechanisms for private sector service delivery can help to transform contact coverage into effective coverage.

Previous research has shown that relying only on facility-based health service delivery may not be appropriate for preventive service delivery in Bangladesh. In the current policy setup, only a subset of children is reached by services provided during sick child visits and coverage of ANC for women continues to be sub-optimal.[16] To improve overall service coverage and quality, there is a need to strengthen existing services, while also exploring other avenues to improve access and coverage of key nutrition interventions through new service delivery contacts. Potential approaches include the provision of nutrition interventions in a preventative setting or via outreach activities, including home visits by front-line workers, community-based events for nutrition service interventions during pregnancy, or well-child services held at the community level. Evidence exists about the use of community-based

front-line workers to support infant and young child feeding practices and maternal nutrition in multiple countries including Bangladesh.[44–47] These, and other options, should continue to be explored in Bangladesh to ensure broad scale coverage of high quality nutrition interventions for women and children.

Our study has several strengths. Studies using data from household surveys often report contact coverage estimates that are limited by the respondent recall, but do not capture the content of these services nor their quality. Our study has reported both the services readiness (which was estimated from health facility data), and the contact coverage (which was drawn from household survey data) and combined these two to estimate input-adjusted coverage. The household surveys and facility data were collected in the same year, thus allowing us to compare facility readiness and services received at the same time for child health services. These data were combined using valid ecological linking method[28] by using facility type and disaggregating by divisions and urban/rural location. Variance and precision of coverage indicators were derived from linked data using the recommended delta method.[29] By including nutrition interventions for multiple populations (urban vs rural; varying levels of education and wealth quintile) in our analysis, we provide a potential tool to assess the health promotion

capacity of the health system by highlighting underserved populations.

We acknowledge some of the limitations of this analysis. First, although we included various facility levels, the exclusion of small private clinics or hospitals with less than 20 beds from the SPA could lead to an overestimation of quality for private sector users as larger facilities tend to have higher readiness than smaller ones. Second, contact coverage was estimated based on women's recall; however, women may not remember exactly where they received services, particularly if they went to more than one type of facility. We have addressed this by restricting the recall period of 3 years for ANC and birth to minimise the recall error. Third, the assumption of no systematic difference in facility readiness within region by the education or socioeconomic stratifies may affect inequality estimates. In reality, differences in facilities in the same category, discrimination in provision care of underserved groups, and lack of individual capacity to extract value from the healthcare system may lead to systematic differences in quality experienced by women of different socioeconomic groups in the same tier of facilities,[48] leading to higher inequalities than are shown here. Fourth, the indicator used here will underestimate contact coverage of growth monitoring services if well children are frequently brought to health facilities for this purpose. Finally, although the SPA survey is quite comprehensive, it does not capture every aspect to measure facility readiness to provide nutrition interventions (such as the infrastructure to implement kangaroo mother care, calcium or food supplements for pregnant women, or IFA and food supplements for children), nor do the data available allow calculation of the care cascade from need for services through health benefits, including quality-adjusted coverage, user adherence-adjusted coverage, or outcome-adjusted coverage.[5 9] Additional data on beneficiary side such as client observations, exit interviews or client perceptions are needed to understand the flow of services provided, services received and whether they can translate into changes in behaviour or practices.

## CONCLUSION

SDG 3 calls for achieving universal health coverage, which requires successfully covering underserved populations with healthcare services as well as ensuring that the quality of services is adequate. Nutrition interventions should be an integral part of health services for mothers and children; however, we find that nutrition input-adjusted coverage was suboptimal and varied subnationally and across the continuum of care in Bangladesh. Special efforts are needed to improve the reach as well as the quality of health and nutrition services to achieve the SDGs.

**Author affiliations**
[1]Poverty, Health and Nutrition Division, International Food Policy Research Institute, Washington, District of Columbia, USA
[2]Hanoi University of Public Health, Hanoi, Viet Nam
[3]Maternal and Child Health Division, ICDDRB, Dhaka, Dhaka District, Bangladesh
[4]Bill and Melinda Gates Foundation, Seattle, Washington, USA
[5]Global Health and Population, Harvard TH Chan School of Public Health, Boston, Massachusetts, USA

**Contributors** PHN conceived the manuscript, conducted the statistical analysis, wrote significant sections of the manuscript and revised the manuscript. QLK conducted the statistical analysis, prepared tables and figures for the manuscript. PP conducted the literature review, wrote significant sections of the manuscript and revised the manuscript. SMB supported data interpretation, reviewed and edited the manuscript. PM supported data interpretation, reviewed and edited the manuscript. EP supported data interpretation, reviewed and edited the manuscript. HL conceived the manuscript, advised on statistical analysis, supported data interpretation and revised the manuscript. All authors read and approved the final submitted manuscript.

**Funding** Bill & Melinda Gates Foundation through POSHAN, led by International Food Policy Research Institute. Grant number: OPP50838.

**Map disclaimer** The depiction of boundaries on this map does not imply the expression of any opinion whatsoever on the part of BMJ (or any member of its group) concerning the legal status of any country, territory, jurisdiction or area or of its authorities. This map is provided without any warranty of any kind, either express or implied.

**Competing interests** None declared.

**Patient consent for publication** Not required.

**Ethics approval** The survey and its procedures were properly reviewed and approved by the authority of the National Institute of Population Research and Training (NIPORT) of the Ministry of Health and Family Welfare. This study used secondary data analyses; thus, ethics approval was not required.

**Provenance and peer review** Not commissioned; externally peer reviewed.

**Data availability statement** Data are available in a public, open access repository. All data relevant to the study are included in the article or uploaded as online supplemental information. Additional original data can be found at the DHS website https://dhsprogram.com/data/available-datasets.cfm.

**ORCID iDs**
Phuong Hong Nguyen http://orcid.org/0000-0003-3418-1674
Hannah H Leslie http://orcid.org/0000-0002-7464-3645

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
