## [Reviewer comments · BMJ Open]

ARTICLE DETAILS

TITLE (PROVISIONAL)	Effective coverage of nutrition interventions across the continuum of care in Bangladesh: Insights from nationwide cross-sectional household and health facility surveys
AUTHORS	Nguyen, Phuong Hong; Khương, Quỳnh Long; Pramanik, Priyanjana; Billah, Sk; Menon, Purnima; Piwoz, Ellen; Leslie, Hannah

VERSION 1 – REVIEW

REVIEWER	Mariame Ouedraogo The SickKids Centre for Global Child Health, Canada
REVIEW RETURNED	02-Jul-2020

GENERAL COMMENTS	Thanks to the authors for submitting this study. The different sections of the manuscript are well elaborated. I commend the authors for a very clear description of the data analysis methods. This assessment was much needed; there is a significant focus on improving contact coverage in LMICs when it is as important to look at those health indicator coverages in the context of health facility capacity and readiness, and quality of services. Having data from exit interviews would have supplemented well your conclusions, but I imagine that information was not available from the 2014 Bangladesh SPA. Perhaps, the authors could consider sharing in the discussion section, their thoughts on how incorporating clients' perceptions is important and would have affected their results/conclusions. There are a few missing punctuation marks (commas, quotation) here and there. For example, it seems there is a comma missing on page 174 between deworming and anemia. I suggest specifying the type of coverage - contact - in table 3 (2nd column) I suggest to the authors to add some additional means of reaching ANC and PNC non-users. Perhaps consider citing some studies/interventions that successfully improved the coverage (contact and effective) of these MCH indicators.
--

REVIEWER	Shannon King Johns Hopkins Bloomberg School of Public Health, USA
REVIEW RETURNED	30-Jul-2020

GENERAL COMMENTS	I appreciate the opportunity to review the manuscript, Effective coverage of nutrition interventions across the continuum of care: Insights from household and health facility data in Bangladesh. The authors adjusted contact-based health coverage estimates, taking into consideration the inputs required to deliver quality nutrition services, across the continuum of care in Bangladesh.
--

Overall, it is a very interesting manuscript and great to see an application of input-adjusted coverage to nutrition interventions. While I think the overall findings are interesting and important, I think there are some areas that could be more thoroughly presented to build on the strength of this study. Please see my detailed comments in the attached document.

MAJOR COMMENTS:

- Broadly speaking in the article there are many different types of 'coverage' measurement mentioned (contact coverage, input-adjusted coverage, quality adjusted coverage, effective coverage). In the introduction (line 75) you define effective coverage in general however you might want to consider similarly defining your other coverage measures succinctly here to ensure all readers can distinguish between all these indicators. Furthermore, I would consider changing the wording to ensure that you use the same terminology throughout all the text to ensure clarity of the measures, for example sometimes contact coverage is just referred to as coverage. Along similar lines, the nutrition interventions are sometimes referred to as interventions and in other instances services, I would suggest keeping one term to ensure it is clear the interventions you are referring to.
- Observation of activities occurring at the time of the visit: The inclusion of these items is a bit strange as these would generally not be considered part of "readiness" and speak more to the provision of care. These are also not part of the SARA indicators so that should be clarified. Can you explain the inclusion of these items in a readiness score? It also appears that some of these items are observed services delivered (i.e. part of provision of care) and others are reported service availability (e.g. Iron supplementation for pregnant women). The latter does not require observation in the SPA tool. There is also overlap between the items in the readiness measures and these specific service offerings. For example, for ANC, IFA tablets are included as a medicine/commodity and then IFA supplementation as a service regularly offered. I'd suggest making sure there isn't duplication across the readiness and provision/availability indicators included in the composite measure if you decide to continue to include the provision/availability indicators.
- There is limited information on how the nutrition interventions and associated readiness items were selected. More detail is needed to justify these choices and to explain whether these are comprehensive measures or are potentially lacking some information to be comprehensive measures of nutrition service quality for pregnant women and children.
- The discussion reiterates the point that coverage and quality need to be improved. However, making the statement to increase inputs is not particularly helpful. Governments in LMICs have difficult decisions to make on how to improve health services with the resources they have available. Is there a specific recommendation to focus on ensuring a specific set of commodities are available? Or to really focus on health worker training and supervision? How can LMICs operationalize the guidance further than just "do better"?

- The limitations don't include some of the fundamental limitations with the SPA data in terms of what has been collected. I would suggest including a mention of the limitations of the SPA survey in capturing readiness to provide nutrition interventions within your limitation section. Also, why didn't you estimate all the steps in the EC cascade? Likely due to data availability but it would be helpful to understand the limitation of the data to have to stop at input-adjusted quality.

- The statistical analysis for generating input-adjusted nutrition measures is lacking details on how the measures were generated. What methodological decisions were taken? How did you handle women for example who sought care at multiple facilities? Did you have any cases in which there was a care-seeking episode from HH survey for which the SPA stratum is empty (i.e. woman obtained ANC from private clinic in region x, but SPA did not sample a private clinic in that region). If yes, how were those handled? What was the rationale for choosing a simple average of items to create the readiness score? Some of this is in the limitations section but should be brought forward to the methods section.

- There writing needs a review for tense shifts as there is variable use of tenses which makes to the flow of the paper a bit difficult to follow.

MINOR COMMENTS:

Here are some specific line by line suggestions for more minor clarifications:

- Acronyms: BEmOC and CEmOC should be BEmONC and CEmONC if the definition includes neonates

- Line 41 in the abstract: It speaks about continuum of care however only prenatal, delivery and child under 5 were examined. I would suggest including a qualification to the continuum of care to ensure it is clear that it refers to prenatal and children under 5 only.

- Line 86-90: I would suggest defining all these stages in brackets as you have done for quality-adjusted coverage and outcome-adjusted coverage to improve clarity.

- Line 134: I suggest expanding upon your explanation of the SPA survey to ensure that the reader is aware that it uses tracer indicators and that many of the items that would be needed to fully capture your selected nutrition interventions are not available within the data. This might help with questions later on for example why within growth monitoring are there no items related to taking and plotting child height.

- Line 159 to 163: it is unclear to me how the definition for coverage of growth monitoring services and coverage of sick-child care differ. I would suggest clarifying, as they appear to differ significantly in terms of contact coverage estimates presented in line 220-221.

- Line 172-173: You mention that the intervention during birth is to support early breastfeeding practices but then in the

	readiness score you have more interventions including vitamin K, KMC, maternal vitamin A.  • Line 173: The postnatal care protocols referred to- are they the national guidance? Or global postnatal guidance? In particular, I am curious as to how this related to vitamin A supplementation and overall, the inclusion of vitamin A supplementation within this analysis as vitamin A is frequently distributed within campaigns rather than provided to children at the health facility. • Line 183 (Table 1): Readiness items are missing detail on the equipment, medicines, diagnostic capacity. This level of detail is important to know which items specifically went into the score as the approach was a simple average of items. • For nutrition interventions during child sick care, you have included ANC guidelines. Is this perhaps a typo? If not, can you explain why ANC guidelines would be relevant to this service? • Line 250 (Table 2): is the third column “% of facilities with item availability”? Above it is mentioned that equal weighting is applied to create the readiness score so it’s unclear if there is a different weighting being applied or if this is referring to using the complex survey design. Also, another small typo- the item says “acid folic” which I think is meant be folic acid. Lastly albendazole and mebendazole are included as separate items however shouldn’t facilities receive readiness credit if they have either of those products as they do the same thing? I have a similar concern for zinc tablet and zinc sulphate syrup as either can be provided and achieve the same result. • Line 258: This sentence mentions delivery care and postnatal care. However, the above indicators don’t differentiate between delivery and post-natal care. Should one of these be related to child health services? • Figure 1: It would be helpful to include confidence intervals in these graphs to better understand if the performance is different between regions and facility types. • Figure 2: What do the colors of the circles mean? Can you add a legend so that this is more informative? • Line 312: Priority of four ANC contacts – WHO has shifted recommendations to a minimum of 8 contacts. Has Bangladesh adopted this new ANC model? • Line 380: This line states “thus bringing our estimates closer into the spectrum of effective coverage”. What does this mean? Confusing with the terminology being presented. • Line 381: You state the timing of the data being collected in the same year is a strength. That is perhaps true for child health services, but not for ANC services as in the coverage survey women are reporting on a birth in the last two years thus ideally for ANC the HFA would be two years prior to the HH survey. • Supplementary Table 2 and 3 are missing many of the medicines for sick child care as compared to what is presented in
--	---

	Table 2 (i.e. Albendazole, mebendazole, iron tablet, vitamin a, zinc tablet, zinc sulphate syrup).
REVIEWER	Hina Khalid Information Technology University, Pakistan.
REVIEW RETURNED	12-Aug-2020
GENERAL COMMENTS	The manuscript is well written and addresses an important and issue. The authors have done a comprehensive job in terms of the literature review and analyses, and the visual aids are especially appealing. The findings have practical implications and can be used for improving the quality and coverage of health interventions.

VERSION 1 – AUTHOR RESPONSE

Reviewer: 1

Thanks to the authors for submitting this study. The different sections of the manuscript are well elaborated. I commend the authors for a very clear description of the data analysis methods. This assessment was much needed; there is a significant focus on improving contact coverage in LMICs when it is as important to look at those health indicator coverages in the context of health facility capacity and readiness, and quality of services.

Response: Thank you for the reviewer's encouraging comments on the importance and strength of our paper.

Having data from exit interviews would have supplemented well your conclusions, but I imagine that information was not available from the 2014 Bangladesh SPA. Perhaps, the authors could consider sharing in the discussion section, their thoughts on how incorporating clients' perceptions is important and would have affected their results/conclusions.

Response: Reviewer is correct that data from exit interviews were not available from the 2014 Bangladesh SPA. We have included some comments in the discussion on the importance of client perceptions and its influences on our results and conclusions (lines 438-446).

There are a few missing punctuation marks (commas, quotation) here and there. For example, it seems there is a comma missing on page 174 between deworming and anemia.

Response: We have fixed the typo as suggested.

I suggest specifying the type of coverage - contact - in table 3 (2nd column)

Response: We have revised second column in Table 3 as suggested.

I suggest to the authors to add some additional means of reaching ANC and PNC non-users. Perhaps consider citing some studies/interventions that successfully improved the coverage (contact and effective) of these MCH indicators.

Response: We have added a paragraph discussing on some potential outreach approach using community health workers or volunteer to provide nutrition services, and also citing interventions that successfully improved the contact coverage and behaviors related to maternal nutrition and infant and young child feeding practices in multiple countries including Bangladesh (lines 399-412).

Reviewer: 2

I appreciate the opportunity to review the manuscript, Effective coverage of nutrition interventions across the continuum of care: Insights from household and health facility data in Bangladesh. The authors adjusted contact-based health coverage estimates, taking into consideration the inputs required to deliver quality nutrition services, across the continuum of care in Bangladesh. Overall, it is

a very interesting manuscript and great to see an application of input-adjusted coverage to nutrition interventions.

Response: Thank you for the reviewer's positive comments on our manuscript.

While I think the overall findings are interesting and important, I think there are some areas that could be more thoroughly presented to build on the strength of this study. Please see my detailed comments in the attached document.

MAJOR COMMENTS:

Broadly speaking in the article there are many different types of 'coverage' measurement mentioned (contact coverage, input-adjusted coverage, quality adjusted coverage, effective coverage). In the introduction (line 75) you define effective coverage in general however you might want to consider similarly defining your other coverage measures succinctly here to ensure all readers can distinguish between all these indicators. Furthermore, I would consider changing the wording to ensure that you use the same terminology throughout all the text to ensure clarity of the measures, for example sometimes contact coverage is just referred to as coverage. Along similar lines, the nutrition interventions are sometimes referred to as interventions and in other instances services, I would suggest keeping one term to ensure it is clear the interventions you are referring to.

Response: We have now defined other coverage measures in lines 88-92 to improve clarity. We have also revised the text, using the same terminology for consistency, and defining the nutrition interventions of interest in Table 1.

Observation of activities occurring at the time of the visit: The inclusion of these items is a bit strange as these would generally not be considered part of "readiness" and speak more to the provision of care. These are also not part of the SARA indicators so that should be clarified. Can you explain the inclusion of these items in a readiness score? It also appears that some of these items are observed services delivered (i.e. part of provision of care) and others are reported service availability (e.g. Iron supplementation for pregnant women). The latter does not require observation in the SPA tool. There is also overlap between the items in the readiness measures and these specific service offerings. For example, for ANC, IFA tablets are included as a medicine/commodity and then IFA supplementation as a service regularly offered. I'd suggest making sure there isn't duplication across the readiness and provision/availability indicators included in the composite measure if you decide to continue to include the provision/availability indicators.

Response: We appreciate this comment and have taken the opportunity to revise and clarify the index we used. Given the fact that the Bangladesh SPA does not include direct observation of clinical care visits, we used self-report or observation of activities at the time of the visit mainly to confirm the routine availability of nutrition interventions. We considered this as one domain and combined with structural input items to create the readiness score. Some items are reported in 2 parts because they compliment for each other. For examples, IFA tablets were available in the medicine domains, but they had no meaning if they only stay on the shelf and were not part of routine service delivery to the clients. Therefore, the availability of IFA routine service provision is important to include. We have revised some indicators in Table 2 (combining IFA and folic acid, albendazole and mebendazole, zinc tablet and zinc sulphate syrup) to avoid the duplication.

There is limited information on how the nutrition interventions and associated readiness items were selected. More detail is needed to justify these choices and to explain whether these are comprehensive measures or are potentially lacking some information to be comprehensive measures of nutrition service quality for pregnant women and children.

Response: We have revised Table 1 (added a column of associated nutrition interventions) to clarify the nutrition interventions that informed selection of readiness items. We have also highlighted the potential lack of some information to be comprehensive measures of nutrition service quality in the discussion (lines 438-446).

The discussion reiterates the point that coverage and quality need to be improved. However, making the statement to increase inputs is not particularly helpful. Governments in LMICs have difficult decisions to make on how to improve health services with the resources they have available. Is there a specific recommendation to focus on ensuring a specific set of commodities are available? Or to really focus on health worker training and supervision? How can LMICs operationalize the guidance further than just "do better"?

Response: We agree with the reviewer's concern that LMIC have to make difficult choices on priorities. This is why the approach we have taken, focusing on quality as well as coverage, is important as it helps improve the return on investment, ensuring that greater impact is achieved for dollar spent. In the discussion, we point to specific actions that can be taken at each contact point to close what Heidkamp et al (2020) referred to as the nutrition opportunity gap. There are steps that can be taken that pertain to provider capacity, commodities, and diagnostic tools, and these vary depending on the type of contact and context. Not all actions will be needed at every-time point, but our text describes that types of things that should be considered. It is not intended to be prescriptive but rather to illustrate the ways that health services and systems can be strengthened to support improved nutrition outcomes (lines 385-398).

We also added a paragraph discussing that in the context of Bangladesh, beyond the health facility, it is important to explore other *outreach* avenues (home visits by community health workers) to improve access and coverage of key nutrition interventions (lines 399-412).

Ref: Heidkamp RA, Wilson E, Menon P, Kuo H, Walton S, Gatica-Dominguez G, Crochemore da Silva I, Aung T, Hajeerhoy N, Piwoz E. How can we realise the full potential of health systems for nutrition? BMJ 2020; 368:16911.

The limitations don't include some of the fundamental limitations with the SPA data in terms of what has been collected. I would suggest including a mention of the limitations of the SPA survey in capturing readiness to provide nutrition interventions within your limitation section. Also, why didn't you estimate all the steps in the EC cascade? Likely due to data availability but it would be helpful to understand the limitation of the data to have to stop at input-adjusted quality.

Response: We have added the limitation of SPA data on capturing facility readiness to provide nutrition interventions, as well as its limitation in tracing all the cascade of effective coverage in lines 438-446.

The statistical analysis for generating input-adjusted nutrition measures is lacking details on how the measures were generated. What methodological decisions were taken? How did you handle women for example who sought care at multiple facilities? Did you have any cases in which there was a care seeking episode from HH survey for which the SPA stratum is empty (i.e. woman obtained ANC from private clinic in region x, but SPA did not sample a private clinic in that region). If yes, how were those handled? What was the rationale for choosing a simple average of items to create the readiness score? Some of this is in the limitations section but should be brought forward to the methods section.

Response: Thank you for noting this; we have provided additional details on how input-adjusted measures were generated. For the readiness score, we re-analyzed the data using the weighted additive scores. Although previous literatures suggested that three methods (simple additive, weighted additive, or PCA approach) yields generally comparable results (Sheffel *et al.*, 2019), we used the recommended weighted additive method (Mallick *et al.*, 2019) which it is easy to calculate and interpret, and reduces the relative importance of variables within a domain while equally weighting domains. We have now added the detailed methods in lines 182-188. For the coverage, we did not have any cases from household for which the SPA stratum is empty. In case women sought ANC care at multiple facilities, we used the highest level of facilities for analyses (lines 199-200).

Mallick, L., Temsah, G. & Wang, W. 2019. Comparing summary measures of quality of care for family planning in Haiti, Malawi, and Tanzania. PLoS One, 14, e0217547.

Sheffel, A., Zeger, S., Heidkamp, R. & Munos, M. K. 2019. Development of summary indices of antenatal care service quality in Haiti, Malawi and Tanzania. BMJ Open, 9, e032558.

There writing needs a review for tense shifts as there is variable use of tenses which makes to the flow of the paper a bit difficult to follow.

Response: We have revised the writing to use consistent past tenses throughout the paper.

MINOR COMMENTS:

Here are some specific line by line suggestions for more minor clarifications:

Acronyms: BEmOC and CEmOC should be BEmONC and CEmONC if the definition includes neonates

Response: We have revised the acronyms as suggested.

Line 41 in the abstract: It speaks about continuum of care however only prenatal, delivery and child

under 5 were examined. I would suggest including a qualification to the continuum of care to ensure it is clear that it refers to prenatal and children under 5 only.

Response: We have revised abstract as suggested (line 41)

Line 86-90: I would suggest defining all these stages in brackets as you have done for quality-adjusted

coverage and outcome-adjusted coverage to improve clarity.

Response: We have now defined all the stages in brackets as suggested (lines 88-91).

Line 134: I suggest expanding upon your explanation of the SPA survey to ensure that the reader is aware that it uses tracer indicators and that many of the items that would be needed to fully capture your selected nutrition interventions are not available within the data. This might help with questions later on for example why within growth monitoring are there no items related to taking and plotting child height.

Response: We have expanded the explanation of the SPA survey in lines 136-137 and 146-147.

Line 159 to 163: it is unclear to me how the definition for coverage of growth monitoring services and coverage of sick-child care differ. I would suggest clarifying, as they appear to differ significantly in terms of contact coverage estimates presented in line 220-221.

Response: growth monitoring activities are intended to be a core part of the national IMCI protocol, thus we assumed all children visit health facility would get growth monitoring services.

The denominators for coverage of growth monitoring services and coverage of sick-child care differ; the former on used all children <5y and the latter used children with illness.

$$\begin{aligned} \text{Coverage of growth monitoring} &= \frac{\text{No of children who sought treatment at a formal health facility}}{\text{All children under 5 years}} *100 \\ \text{Coverage of sick child care} &= \frac{\text{No of sick children who sought treatment at a formal health facility}}{\text{No of children will any illness}} *100 \end{aligned}$$

We have clarified these definitions in lines 164-167.

Line 172-173: You mention that the intervention during birth is to support early breastfeeding practices but then in the readiness score you have more interventions including vitamin K, KMC, maternal vitamin A.

Response: The KMC is important for supporting early breastfeeding practices. We have now revised analyses, omitting vitamin K and maternal vitamin A.

Line 173: The postnatal care protocols referred to- are they the national guidance? Or global postnatal guidance? In particular, I am curious as to how this related to vitamin A supplementation and overall, the inclusion of vitamin A supplementation within this analysis as vitamin A is frequently distributed within campaigns rather than provided to children at the health facility.

Response: This is national guidance. Vitamin A is normally distributed within campaigns, but should also be available at health facilities and provided for those who sought the services- therefore it was included.

Line 183 (Table 1): Readiness items are missing detail on the equipment, medicines, diagnostic capacity. This level of detail is important to know which items specifically went into the score as the approach was a simple average of items.

Response: We have provided additional information on the equipment, medicines, diagnostic capacity in Table 1.

For nutrition interventions during child sick care, you have included ANC guidelines. Is this perhaps a typo? If not, can you explain why ANC guidelines would be relevant to this service?

Response: Sorry it is a typo. We have revised the text to IMCI guidelines.

Line 250 (Table 2): is the third column “% of facilities with item availability”? Above it is mentioned that equal weighting is applied to create the readiness score so it’s unclear if there is a different weighting being applied or if this is referring to using the complex survey design. Also, another small typo- the item says “acid folic” which I think is meant to be folic acid. Lastly albendazole and mebendazole are included as separate items however shouldn’t facilities receive readiness credit if they have either of those products as they do the same thing? I have a similar concern for zinc tablet and zinc sulphate syrup as either can be provided and achieve the same result.

Response: We have clarified the title of the third column in Table 2 (change to “% facilities with item available” as suggested). We also provided more information on the weighted additive method in the method section (lines 182-188). In Table 2, we have combined IFA and folic acid, albendazole and mebendazole, zinc tablet and zinc sulphate syrup to avoid the duplication.

Line 258: This sentence mentions delivery care and postnatal care. However, the above indicators don’t differentiate between delivery and post-natal care. Should one of these be related to child health services?

Response: We have revised the sentence to clarify that the later indicator is related to child health services (lines 274).

Figure 1: It would be helpful to include confidence intervals in these graphs to better understand if the performance is different between regions and facility types.

Response: We have included confidence intervals in Figure 1 as suggested.

Figure 2: What do the colors of the circles mean? Can you add a legend so that this is more informative?

Response: the colors of the circles represent magnitude of coverage where the red one shows low coverage and green one shows higher coverage. We have added a legend to the figure 2 as suggested.

Line 312: Priority of four ANC contacts – WHO has shifted recommendations to a minimum of 8 contacts. Has Bangladesh adopted this new ANC model?

Response: The current practice of the ANC program in Bangladesh follows the earlier WHO’s guidelines of 4+ ANC visits. The government of Bangladesh along with nongovernmental and international organizations are working together to increase the number of ANC contacts, but only 8% of women achieved 8+ ANC visits (Chanda *et al.*, 2020).

Chanda, S. K., Ahammed, B., Howlader, M. H., Ashikuzzaman, M., Shovo, T. E. & Hossain, M. T. 2020. Factors associating different antenatal care contacts of women: A cross-sectional analysis of Bangladesh demographic and health survey 2014 data. PLoS One, 15, e0232257.

Line 380: This line states “thus bringing our estimates closer into the spectrum of effective coverage”. What does this mean? Confusing with the terminology being presented.

Response: We have deleted this text to avoid confusion.

Line 381: You state the timing of the data being collected in the same year is a strength. That is perhaps true for child health services, but not for ANC services as in the coverage survey women are reporting on a birth in the last two years thus ideally for ANC the HFA would be two years prior to the HH survey.

Response: We have specified in the discussion that the strength of data collection in the same year is applied for child health services only (lines 419-420).

Supplementary Table 2 and 3 are missing many of the medicines for sick child care as compared to what is presented in Table 2 (i.e. Albendazole, mebendazole, iron tablet, vitamin a, zinc tablet, zinc sulphate syrup).

Response: We have added the missing information for the medicines related to sick child care in Supplementary Table 2 and 3 as suggested.

Reviewer: 3

The manuscript is well written and addresses an important and issue. The authors have done a comprehensive job in terms of the literature review and analyses, and the visual aids are especially appealing. The findings have practical implications and can be used for improving the quality and coverage of health interventions.

Response: Thank you for your positive comments on our paper.

VERSION 2 – REVIEW

REVIEWER	Mariame Ouedraogo Centre for Global Child Health, The Hospital for Sick Children, Canada
REVIEW RETURNED	12-Oct-2020

GENERAL COMMENTS	No additional comments for the authors
--

REVIEWER	Shannon King Johns Hopkins Bloomberg School of Public Health, USA
REVIEW RETURNED	12-Oct-2020

GENERAL COMMENTS	I appreciate the opportunity to review the updated version of the manuscript, Effective coverage of nutrition interventions across the continuum of care in Bangladesh: Insights from nationwide cross-sectional household and health facility surveys. Thanks to the authors for the revisions that were made based on the feedback from reviewers. Overall I believe the paper is a great application of input-adjusted coverage to nutrition interventions but there are still some comments that I believe need to be addressed. Major comments:  1. The inclusion of availability items within the readiness score is still a point of concern. Firstly, the availability of the services should only be included if it is utilized as the denominator to identify which health care centers should be assessed for readiness rather than included within a readiness score. Depending on the type of health facility, they may or not may not have that service included within their scope of care. Secondly, the items that were utilized to assess availability are described as either “self-reported or observation of activities at the time of visit”, these items are truly indicators of service availability and service provision and not service readiness. Third, using the service provision items as a proxy measures for availability of services is problematic as many clinics in theory offer a service however whether or not that service is actually provided on any one given day is highly variable. Finally,, there are still concerns with the duplication of items when you include these service availability/provision of care items as it results in differential weighting of the nutrition interventions for each of the age groups. I would suggest removing the services availability and provision of care items from the indices. 2. It is unclear when the author refers to “Iron-folic acid supplements” if they have available IFA only as a combination tablet, or if facilities given credit if they had both iron and folic acid available in separate tablets? For example, the first item in Table 2 is “IFA/folic acid supplementation” but these two services are not
---

	equivalent. Additionally, for the Medicine in Table 2, the item is iron or folic acid tablets, these are not the equivalent so how are these included in one item only? The correct items should be included within the index and the text should be revised to ensure it is clear what item and service is being referred to. Iron supplementation can be offered as a stand-alone intervention and IFA can be offered as a combined intervention to address different health issues. Folic acid supplementation during pregnancy alone is not an evidence-based intervention. A few minor comments for the authors:  1. Line 111: I think it would be helpful to include the 4 nutrition interventions you reference so they are clearly defined here. 2. Line 136-147: Previous comment suggested that you explicitly mention that the SPA survey uses tracer indicators rather than assessing all items that would be required to deliver the intervention. The adjusted text still does not reflect this. I would encourage you to ensure that the language of tracer indicators are included because there are evident gaps in the nutrition interventions included in your analysis compared to those that are provided during pregnancy as many of the pregnancy items do not have tracer indicators within the SPA survey. Additionally, elaborating on the tracer indicators would help with explaining why there aren't some of the items for the interventions within the readiness indices, for example KMC requires significant equipment if it is implemented according to the guidance documents. 3. Line 164-167: There are still some challenges with the definition of contact coverage of growth monitoring. The numerator for coverage of growth monitoring should include all children (sick or not sick) meaning that it should be "No of children who sought care at a formal health facility" rather than treatment because they are not necessarily seeking treatment. Secondly, for this numerator, there is currently only data available for those seeking treatment (ie those sick children). There is no information available for the prevalence of healthy children who received growth monitoring. Therefore, your coverage of growth monitoring is likely a relatively large underestimation of those who are actually receiving growth monitoring. 4. Figure 1: It might be helpful to define the acronyms within your labels. 5. Figure 2: Is the y-axis referring to contact coverage? If so, it might be helpful to include this to improve clarity.
--	--

VERSION 2 – AUTHOR RESPONSE

Reviewer: 1

No additional comments for the authors

Reviewer: 2

I appreciate the opportunity to review the updated version of the manuscript, Effective coverage of nutrition interventions across the continuum of care in Bangladesh: Insights from nationwide cross-sectional household and health facility surveys. Thanks to the authors for the revisions that were made based on the feedback from reviewers. Overall I believe the paper is a great application of input-adjusted coverage to nutrition interventions but there are still some comments that I believe need to be addressed.

Major comments:

1. The inclusion of availability items within the readiness score is still a point of concern. Firstly, the availability of the services should only be included if it is utilized as the denominator to identify which health care centers should be assessed for readiness rather than included within a readiness score. Depending on the type of health facility, they may or not may not have that service included within their scope of care. Secondly, the items that were utilized to assess availability are described as either “self-reported or observation of activities at the time of visit”, these items are truly indicators of service availability and service provision and not service readiness. Third, using the service provision items as a proxy measures for availability of services is problematic as many clinics in theory offer a service however whether or not that service is actually provided on any one given day is highly variable. Finally,, there are still concerns with the duplication of items when you include these service availability/provision of care items as it results in differential weighting of the nutrition interventions for each of the age groups. I would suggest removing the services availability and provision of care items from the indices.

Response: We have removed the services availability and provision of care items from the indices, rerun the analyses, revised Tables/figures and correspondent text accordingly.

2. It is unclear when the author refers to “Iron-folic acid supplements” if they have available IFA only as a combination tablet, or if facilities given credit if they had both iron and folic acid available in separate tablets? For example, the first item in Table 2 is “IFA/folic acid supplementation” but these two services are not equivalent. Additionally, for the Medicine in Table 2, the item is iron or folic acid tablets, these are not the equivalent so how are these included in one item only? The correct items should be included within the index and the text should be revised to ensure it is clear what item and service is being referred to. Iron supplementation can be offered as a stand-alone intervention and IFA can be offered as a combined intervention to address different health issues. Folic acid supplementation during pregnancy alone is not an evidence-based intervention.

Response: We have now clarified two items: iron tablets and IFA tablets in Table 1 and Table 2.
Response:

A few minor comments for the authors:

1. Line 111: I think it would be helpful to include the 4 nutrition interventions you reference so they are clearly defined here.

Response: We have now included the 4 nutrition interventions in lines 111-112 as suggested.

2. Line 136-147: Previous comment suggested that you explicitly mention that the SPA survey uses tracer indicators rather than assessing all items that would be required to deliver the intervention. The adjusted text still does not reflect this. I would encourage you to ensure that the language of tracer indicators are included because there are evident gaps in the nutrition interventions included in your analysis compared to those that are provided during pregnancy as many of the pregnancy items do not have tracer indicators within the SPA survey. Additionally, elaborating on the tracer indicators would help with explaining why there aren't some of the items for the interventions within the readiness indices, for example KMC requires significant equipment if it is implemented according to the guidance documents.

Response: We have revised the paragraph, including the language of tracer indicators in lines 141-142, 176-177. We also expanded on limitations in line 438.

3. Line 164-167: There are still some challenges with the definition of contact coverage of growth monitoring. The numerator for coverage of growth monitoring should include all children (sick or not sick) meaning that it should be “No of children who sought care at a formal health facility” rather

than treatment because they are not necessarily seeking treatment. Secondly, for this numerator, there is currently only data available for those seeking treatment (ie those sick children). There is no information available for the prevalence of healthy children who received growth monitoring. Therefore, your coverage of growth monitoring is likely a relatively large underestimation of those who are actually receiving growth monitoring.

Response: We have clarified the numerator relates to care seeking in line 166. We agree with the challenges of estimating growth monitoring due to unavailable information such as care seeking among well children. We have clarified that the indicator, while incomplete, follows the service delivery model in Bangladesh that has oriented growth monitoring services via IMCI (lines 166-169) and discussed about this indicator in the limitation (lines 434-436).

4. Figure 1: It might be helpful to define the acronyms within your labels.

Response: we have now defined acronyms in the figure legend (lines 458-459).

5. Figure 2: Is the y-axis referring to contact coverage? If so, it might be helpful to include this to improve clarity.

Response: We have clarified the y-axis as contract coverage as suggested.

VERSION 3 – REVIEW

REVIEWER	Shannon King Johns Hopkins School of Public Health
REVIEW RETURNED	15-Nov-2020

GENERAL COMMENTS	Thank you for your revisions and edits to the paper. I would suggest reviewing text line 256-258 as the numbers in the text do not match the numbers in the table. In particular IFA supplements are available in 83% of facilities (Table 2) but the text says 91%. Similarly Vitamin A and zinc are available in 66% of facilities (Table 2) but the text says three quarters of facilities.
--